# Spatiotemporal Variation in Ground Level Ozone and Its Driving Factors: A Comparative Study of Coastal and Inland Cities in Eastern China

**DOI:** 10.3390/ijerph19159687

**Published:** 2022-08-05

**Authors:** Mengge Zhou, Yonghua Li, Fengying Zhang

**Affiliations:** 1Key Laboratory of Land Surface Pattern and Simulation, Institute of Geographic Sciences and Natural Resources Research, Chinese Academy of Sciences, Beijing 100101, China; 2University of Chinese Academy of Sciences, Beijing 100049, China; 3China National Environmental Monitoring Centre, Beijing 100012, China

**Keywords:** ozone, coastal and inland cities, spatiotemporal distribution, wavelet analysis, multiscale geographically weighted regression

## Abstract

Variations in marine and terrestrial geographical environments can cause considerable differences in meteorological conditions, economic features, and population density (PD) levels between coastal and inland cities, which in turn can affect the urban air quality. In this study, a five-year (2016–2020) dataset encompassing air monitoring (from the China National Environmental Monitoring Centre), socioeconomic statistical (from the Shandong Province Bureau of Statistics) and meteorological data (from the U.S. National Centers for Environmental Information, National Oceanic and Atmospheric Administration) was employed to investigate the spatiotemporal distribution characteristics and underlying drivers of urban ozone (O_3_) in Shandong Province, a region with both land and sea environments in eastern China. The main research methods included the multiscale geographically weighted regression (MGWR) model and wavelet analysis. From 2016 to 2019, the O_3_ concentration increased year by year in most cities, but in 2020, the O_3_ concentration in all cities decreased. O_3_ concentration exhibited obvious regional differences, with higher levels in inland areas and lower levels in eastern coastal areas. The MGWR analysis results indicated the relationship between PD, urbanization rate (UR), and O_3_ was greater in coastal cities than that in the inland cities. Furthermore, the wavelet coherence (WTC) analysis results indicated that the daily maximum temperature was the most important factor influencing the O_3_ concentration. Compared with NO, NO_2_, and NO_x_ (NO_x_
**≡** NO + NO_2_), the ratio of NO_2_/NO was more coherent with O_3_. In addition, the temperature, the wind speed, nitrogen oxides, and fine particulate matter (PM_2.5_) exerted a greater impact on O_3_ in coastal cities than that in inland cities. In summary, the effects of the various abovementioned factors on O_3_ differed between coastal cities and inland cities. The present study could provide a scientific basis for targeted O_3_ pollution control in coastal and inland cities.

## 1. Introduction

Ozone (O_3_) is a typical secondary pollutant gas produced via photochemical oxidation reactions [1]. Recently, O_3_ pollution has received profound attention in China. Through various efforts over the past few years, the concentration of fine particulate matter (PM_2.5_) has been effectively controlled in most parts of China, but O_3_ pollution is increasing [2,3]. From 2016 to 2020, the proportion of Chinese cities at the prefecture level and above with the daily maximum 8-h average (DMA8) O_3_ concentration exceeding the national secondary standard (160 μg/m^3^) was 17.5%, 32.3%, 34.6%, 30.6%, and 16.6% [4,5,6,7,8]. Many previous studies have examined O_3_ pollution in the North China Plain, Yangtze River Delta, and Pearl River Delta [9,10,11,12,13,14,15,16,17]. When a certain O_3_ concentration is exceeded, this can exert an adverse impact on the ecological environment, food safety, human health, and climate change [18,19,20,21,22,23,24,25]. Therefore, the Chinese government issued relevant policies in 2020 to promote the prevention and control of O_3_ pollution [26].

Volatile organic compounds (VOCs) and nitrogen oxides (NO_x_ ≡ NO + NO_2_) are precursors of O_3_ [27]. Many studies have explored the relationship, photochemical reaction mechanism, and influence factors between O_3_ and its precursors [28,29,30,31,32,33]. Meteorological conditions are important factors of O_3_ generation and transmission. Radiation enhancement, temperature increase, and sunshine time extension facilitate O_3_ generation, while a high relative humidity reduces O_3_ [3,34,35,36]. In addition to meteorological conditions, there may also be a relationship between socioeconomic factors and the spatial distribution of air pollution [37]. Many studies have analyzed the correlation between PM_2.5_ and socioeconomic factors [38,39]. However, there is little research on the socioeconomic factors of O_3_. Even more notably, few studies have examined the combined impact of the O_3_ precursors, meteorological conditions, and socioeconomic factors on O_3_ pollution from a spatiotemporal perspective. These influencing factors may have different effects in different regions, which is worth exploring.

The difference in thermal conditions between land and oceans results in distinct climates between inland and coastal areas. Coupled with differences in the geographical environment, there could occur a comprehensive effect leading to different economic development and population distributions between inland and coastal areas. Within the context of global warming, the contrast between land and ocean warming levels is increasing (land warming enhanced), which intensifies aerosol pollution [40]. Considering that the influence of various factors on the O_3_ concentration in coastal and inland cities may differ, it is therefore necessary to evaluate these differences.

The multiscale geographically weighted regression (MGWR) model is an extension of the geographically weighted regression (GWR) model, which is commonly used in the analysis of spatial influencing factors [41]. The MGWR model considers that different spatial scales exert different effects of the action mode and intensity on the spatial relationship between the considered factors and dependent variables, so it can explore the spatial heterogeneity of influencing factors [41]. MGWR has been widely used in the environmental field, such as PM_2.5_, PM_10_, NO_2_, and other air pollutants [42,43,44]. Zhan et al. (2022) has examined the spatial heterogeneity effects of both socioeconomic factors and natural factors on continuous air pollution with a MGWR [45]. Thus, we attempted to analyze the underlying drivers affecting the spatial distribution of O_3_ and its spatial heterogeneity from socioeconomic aspects using MGWR.

Wavelet analysis was developed based on the Fourier transform [46]. The wavelet transform extends time series into the time–frequency space to overcome the limitations of the Fourier transform [47]. Wavelet analysis is a powerful analysis tool highly suitable to study nonstationary processes in the finite space–time domain [46,48]. This method has been widely applied in geophysics, economics, and public health research [49,50,51,52,53,54,55,56]. Therefore, this study will explore the potential drivers affecting the spatial distribution of O_3_ from the perspective of meteorological conditions and other air pollutants through wavelet analysis (wavelet coherence (WTC)).

Shandong Province is located in eastern China, bordering the Bohai Sea and Yellow Sea. There occur both typical coastal cities and typical inland cities due to the large east–west span. In addition, Shandong Province ranks third in terms of its economy and second in terms of its population in China. The well-developed economy, large population, and high consumption of resources have caused serious air pollution, including O_3_ pollution, in this area. Based on the above, we reasonably assume that the differences in the socioeconomic and natural environment of Shandong Province lead to significant differences in O_3_ concentrations between coastal and inland areas, and the influence of various factors in different cities is different. This study will first attempt to combine MGWR and wavelet analysis to explore the influence of various factors on O_3_ from multiple perspectives, so as to identify the key factors influencing the O_3_ distribution. This study can provide the Chinese government with a more targeted reference for the treatment of O_3_ pollution in coastal and inland cities.

## 2. Data and Methods

### 2.1. Study Area

Shandong Province is located between 34°22.90′~38°24.01′ N and 114°47.50′~122° 42.30′ E (Figure 1). The region contains a complex, diverse terrain, and a long coastline, with four typical coastal cities: Qingdao, Yantai, Weihai and Rizhao. Shandong Province exhibits four distinct seasons, belonging to the warm temperate monsoon climate. The weather is changeable in spring, with less rain and windy sand-prone conditions. In summer, controlled by the southeast marine monsoon, southerly winds prevail. In autumn, the weather is sunny and moderate. Winter is controlled by the continental monsoon climate, mostly involving northerlies. The mountains in the central part of the territory are raised, which comprise mountains with an altitude higher than a kilometer, such as Mount Tai, Mount Lu, Mount Yi, and Mount Meng. The surrounding areas gradually transition from low mountains and hills into plains.

### 2.2. O_3_ and Other Air Pollutants Data

The DMA8 O_3_ concentration and the 24-h average concentrations of NO, NO_2_, NO_x_, PM_2.5_, and respirable particles (PM_10_) were considered in this study. Data pertaining to 16 cities in Shandong Province from 1 January 2016, to 31 December 2020, were obtained from the China National Environmental Monitoring Centre (CNEMC, http://www.cnemc.cn/, accessed on 13 September 2021). The annual O_3_ concentration is defined as the 90th percentile of DMA8 O_3_ concentration [57]. O_3_ monitoring and recording were carried out in strict accordance with Chinese national standards [58]. The data statistical validity of each evaluation project was implemented in accordance with the relevant provisions of the Chinese national standard [59].

### 2.3. Socioeconomic and Meteorological Data

The socioeconomic data were derived from the statistics, including data on the population, gross domestic product (GDP), industrial power consumption, and number of civil vehicles (NCV) for 16 cities from 2016 to 2020 [60,61,62,63,64]. We also compiled the daily meteorological data from National Oceanic and Atmospheric Administration (NOAA) National Centers for Environmental Information (https://www.ncei.noaa.gov/maps/global-summaries/, accessed on 5 September 2021). The meteorological monitoring sites are located in Jinan (116.9833° E, 36.6833° N) and Qingdao (120.3744° E, 36.2661° N). The specific monitoring items include average wind speed (WDSP, m/s), maximum continuous wind speed (MXSPD, m/s), maximum temperature (MAX, °C), and average temperature (TEMP, °C).

### 2.4. Methods and Mechanism

We used ArcGIS to visualize the spatiotemporal variation characteristics of O_3_ concentration in Shandong Province from 2016 to 2020 and conduct hot spot analysis. OLS analysis was performed in R 4.0.5 invented by Rose Ihaka and Robert Gentleman of New Zealand. The GWR and MGWR models were implemented in MGWR 2.2 provided by the School of Geographical Sciences and Urban Planning at Arizona State University (https://sgsup.asu.edu accessed on 5 September 2021). Wavelet analysis was performed using publicly available MATLAB code.

#### 2.4.1. GWR and MGWR

The GWR model is a new method to incorporate spatial correlation into a regression model, which allows regression coefficients to vary in space [41]. The expression of GWR model is as follows:(1)$$y_{i}=\beta_{0}\left(u_{i},v_{i}\right)+\underset{j}{\sum }\beta_{j}\left(u_{i},v_{i}\right)x_{ij}+\varepsilon_{i}$$
where (*u_i_*, *v_i_*) denotes the spatial coordinates of the *i*th observation point, *β* (*u_i_*, *v_i_*) is the regression coefficient of the *j*th independent variable at the *i*th observation point, *β_0_* (*u_i_*, *v_i_*) is the intercept of the model at the *i*th observation point, and *e_i_* is the error term.

The MGWR model is an extension of the GWR model [41]. The largest difference between these models is the bandwidth. All variables in the GWR model exhibit the same bandwidth. However, the MGWR model specifies a dedicated bandwidth for each variable, thus reducing estimation errors and ensuring a more realistic and useful spatial process model. Different bandwidths can reveal the scale effect of different factors on O_3_ concentration changes. Generally, the larger the bandwidth is, the lower the spatial heterogeneity [44]. The calculation equation is as follows:(2)$$y_{i}=\beta_{0}\left(u_{i},v_{i}\right)+\underset{j}{\sum }\beta_{bwj}\left(u_{i},v_{i}\right)x_{ij}+\varepsilon_{i}$$
where (*u_i_*, *v_i_*) denotes the coordinates of position *i*, *bwj* denotes the bandwidth considered by the regression coefficient of the *j*th variable, and *β_bw j_*(*u_i_*, *v_i_*) is the regression coefficient of the *j*th variable at *i*.

#### 2.4.2. Wavelet Analysis

Wavelet analysis is based on Fourier analysis. Elucidation of the localization characteristics of the analyzed object in the time and frequency domains constitutes the advantage of wavelet analysis [65]. The continuous wavelet transform (CWT) method entails wavelet superposition of different scales and displacement levels [66]. One frequently applied wavelet function is the Morlet wavelet, which is also adopted in this paper. Based on two CWTs, Grinsted et al. (2004) constructed the cross-wavelet transform (XWT), which exposes the high common power and relative phase in time–frequency space [67]. The XWT of two time series *x_n_* and *y_n_* is denoted as *W^XY^,* and the cross-wavelet power is denoted as |*W^XY^*|. The complex argument arg(*W^XY^*) can be interpreted as the local relative phase between *x_n_* and *y_n_* in time–frequency space and is given as follows:(3)$$\phi \left(s,t\right)=\tan^{-1}(Im\left({\overset{↔}{W}}^{y,x}\left(s,t\right)\right)/Re{\overset{↔}{W}}^{y,x}\left(s,t\right))$$
where ${\overset{↔}{W}}^{y,x}\left(s,t\right)$ is the matrix of the smoothed cross-wavelet power spectra between *x_n_* and *y_n_* and *Im* and *Re* denote the imaginary and real parts, respectively, of ${\overset{↔}{W}}^{y,x}\left(s,t\right)$ [51]. The phase angle can be determined to analyze the variation relationship (lag or consistent change) between two time series [49]. Consequently, left- and right-pointing phase angles indicate antiphase and in-phase relationships, respectively.

WTC analysis can be employed to study the correlation between two data series in time–frequency space. The WTC can significantly enhance the linear relationship and determine the covariance intensity between two time series [49,51]. Grinsted et al. (2004) defined the WTC of two time series as follows:(4)$$R_{\;n}^{2}\left(s\right)=\frac{{\left|S\left(s^{-1}W_{n}^{XY}\left(s\right)\right)\right|}^{2}}{S\left(s^{-1}{\left|W_{n}^{X}\left(s\right)\right|}^{2}\right)\cdot S\left(s^{-1}{\left|W_{n}^{X}\left(s\right)\right|}^{2}\right)}$$
where *S* is a smoothing operator and *W^X^* and *W^Y^* are the CWTs of *x_n_* and *y_n_*, respectively. *R*^2^ ranges from 0 to 1.
(5)$$S\left(W\right)=S_{scale}\left(S_{time}\left(W_{n}\left(s\right)\right)\right)$$
where *S_scale_* denotes the smoothing level along the wavelet scale axis and *S_time_* denotes the smoothing level in time.

### 2.5. Choice of Two Typical Cities

Qingdao is the most economically well-developed city in Shandong Province, and Jinan is the capital city of Shandong Province. Furthermore, Qingdao is a coastal city, while Jinan is an inland city (Figure 1). There are great geographical differences between these cities. Thus, this study chose Jinan and Qingdao as typical representative cities to explore the differences of the time–frequency relationship between O_3_ and meteorological factors and other air pollutants between coastal and inland cities.

## 3. Results and Discussion

### 3.1. Spatiotemporal Distribution Characteristics of O_3_

The annual O_3_ concentration changes in 16 cities in Shandong Province are shown in Figure 2. Significant spatiotemporal variations in the O_3_ distribution were observed. Over the past five years, only three cities in Shandong saw a slight decrease in O_3_ concentration in 2020, while the other 13 cities saw an increase. The growth range of the O_3_ concentration in the 13 cities was 2.42~49.60%, with an average increase of 15.40%. From 2016 to 2019, the O_3_ concentration in most cities increased year by year. An important reason for the observed increase could be enhancement in anthropogenic precursors. However, in 2020, the O_3_ concentration was reduced due to strict control measures implemented by the Chinese government and the COVID-19 pandemic. The same trend is also shown in Figure 3, which indicates the annual changes in the O_3_ concentration attainment rate in each city. The standard Ⅰ concentration limit of the DMA8 O_3_ concentration is 100 μg/m^3^, which is consistent with the air quality guideline (AQG) level of the World Health Organization Global Air Quality Guidelines (WHOGAQG) [68]. The standard Ⅱ concentration limit of the DMA8 O_3_ concentration is 160 μg/m^3^, which is consistent with the interim target 1 level of the WHOGAQG [68]. From 2016 to 2020, the proportion of the number of days when the O_3_ concentration reached standard Ⅰ in Shandong Province was 60.55%, 55.92%, 51.61%, 46.61%, and 49.62%, respectively. The number of days when the O_3_ concentration reached standard Ⅱ accounted for 90.83%, 87.00%, 84.47%, 79.95%, and 85.11%, respectively. The most serious O_3_ pollution occurred in 2019, with 53.39% and 21.05% exceeding standard Ⅰ and standard Ⅱ, respectively. Only three coastal cities (Qingdao: 207/365; Rizhao: 188/365; Yantai: 187/365) exhibited more than half of the total number of days with the concentration reaching standard Ⅰ.

Moreover, there were obvious differences in the O_3_ distribution between inland and coastal areas. The O_3_ concentration in the inland areas (central and northwestern Shandong) was higher than that on the Jiaodong Peninsula (mainly coastal cities), which is consistent with Yao et al. (2019) [69]. This study further explored the spatial distribution characteristics of O_3_ through hot spot analysis (Figure 4). O_3_ cold spot areas were located in Qingdao, Yantai and Weihai. In addition, O_3_ high risk areas expanded from Jinan in 2016 to Jinan and surrounding areas in 2020, including Dezhou, Binzhou and Zibo.

### 3.2. Socioeconomic Impacts

In this study, redundant factors were eliminated according to the variance inflation factor (VIF) value (VIF < 7.5) of the ordinary least squares (OLS) model. VIF is a multicollinearity test method. The closer the VIF value is to 1, the lighter the multicollinearity is, and vice versa. Seven factors were selected: time, population density (PD, 10,000 people·hm^−^^2^), urbanization rate (UR, %), proportion of the secondary industry (PSI, %), output value of farming, forestry, animal husbandry and fishery (OFFAF, CNY), industrial power consumption (IPC, Billion kW·h), and NCV (unit). We tried to use PD to describe the population distribution, UR to represent the level of regional economic development, PSI to show the industrial structure, OFFAF to represent the development level of the primary industry, IPC to mean the industrial emissions of ozone, and NCV to represent the motor vehicle emissions of ozone. Three models, namely, OLS, GWR, and MGWR models, were employed to explore the socioeconomic relationship between these factors and the O_3_ concentration.

According to the corrected AICc, *R*^2^ and adj-*R*^2^ values of these three models (Table 1), the MGWR model achieved the best fitting effect. Compared with the OLS model, the *R*^2^ and adj-*R*^2^ values of the GWR and MGWR were greatly improved. But MGWR can more truly reflect the relationship between socioeconomic factors and O_3_ than GWR because the bandwidth of the MGWR model is variable, while the bandwidth of the GWR model is fixed. The MGWR model can reflect the spatial difference of the relationship between the independent variable and the dependent variable.

According to the OLS results (Table 1), there was a significant correlation between Time, PD, UR, PSI, and O_3_. Combined with Figure 2 and Figure 3, the O_3_ concentration had a distinct increasing trend from 2016 to 2020. The effects of PD and PSI on the O_3_ concentration were positive across the whole province. An increase in PD and PSI could lead to an increase in the O_3_ concentration to varying degrees. Densely populated areas contributed more to O_3_ due to the consumption of many resources as fuel [70]. Cities with high industrial proportions produced more O_3_ precursors, resulting in higher O_3_ concentration. In most areas of Shandong Province, especially in coastal areas, there existed a stronger negative correlation between UR and the O_3_ concentration. This is because coastal cities had a high degree of urbanization, but their O_3_ concentration was low because of low emissions and favorable diffusion conditions. OFFAF, IPC, and NCV slightly affected O_3_ and the relationship between them was not significant. Figure 5 indicates that compared with vehicle exhaust emissions, industrial emissions contributed more in the south of Shandong Province. The socioeconomic environment had a certain and extremely complex impact on the formation of O_3_. Greater population density and higher industrialization levels may lead to more O_3_ precursors. In addition, due to the implementation of air pollution prevention and control policies, the areas with higher economic development levels are willing to invest higher more in air pollution control and better treatment effects. Notably, Lee et al. (2021) found that the number of houses, number of parking lots, and area of reconstruction projects were positively correlated with O_3_ rates [71].

In this study, the effects of the mode and intensity of Time, PD, PSI, and NCV on the change in O_3_ concentration were roughly similar on a large spatial scale, and the spatial relationship tended to remain stable (bandwidth is 71). The spatial heterogeneity in the impact of UR, OFFAF, and IPC on the O_3_ concentration was high (bandwidths are 62, 57, and 57 respectively). What is more, Figure 5 shows the relationship between PD, UR, and O_3_ is greater in coastal cities than that in the inland cities.

As a secondary air pollutant, processes leading to variations in surface O_3_ are complex because they are impacted by natural and human factors. The impact of socioeconomic conditions on O_3_ is only a small part, while the meteorological environment has a greater impact on O_3_. Therefore, it is difficult to explain the O_3_ variations simply by socioeconomic statistics. To this end, we used meteorological and other air pollution data in the next section to further analyze its impact on O_3_.

### 3.3. Impact of Other Air Pollutants and Meteorological Factors

#### 3.3.1. Wavelet Power Spectrum of O_3_ in Jinan and Qingdao

The area enclosed by the black line in the Figure 6 indicates that the periodicity within the time series is significant. There occurred an obvious annual cycle in Qingdao and Jinan. The O_3_ concentrations in those two cities frequently fluctuated in summer (June to August) and autumn (September to November). In addition, the Qingdao O_3_ concentration exhibited a moderate cycle of approximately 150 days from 2016 to 2019. This indicates that double peaks occurred in the O_3_ concentration change in Qingdao, one peak in May or June and the other peak in September (Appendix A). Affected by the summer monsoon climate, Qingdao is rainy in July and August. This led to a significant decline in the O_3_ concentration in summer due to lack of light [34,72].

#### 3.3.2. Wavelet Coherence Coefficient

In this study, NO, NO_2_, NO_x_, NO_2_/NO, PM_10_, PM_2.5_, WDSP, MXSPD, TEMP, and MAX were selected to explore the corresponding time–frequency relationship with O_3_. The arithmetic mean value of the WTC coefficient (ARsq) is provided in Table 2. In summary, the variation in O_3_ in Qingdao was more closely related to other air pollutants than that in Jinan. Perhaps it is because Qingdao is greatly affected by the Yellow Sea, unlike Jinan, which is affected by many surrounding cities (Appendix A). The time–frequency relationship between O_3_ and other gas pollutants in Jinan was more complex. Among the meteorological elements, the coherence between the temperature and O_3_ in Jinan was higher than that in Qingdao, while the coherence between the wind speed and O_3_ in Qingdao was relatively higher. Previous studies have reported that O_3_ generation largely depends on high temperatures and strong solar radiation [3,73]. Jinan’s temperature fluctuated more greatly than that in Qingdao because Jinan is located far from the ocean. As a result, O_3_ in Jinan was more vulnerable to the temperature. The wind speed notably impacted O_3_ in Qingdao due to the prevalence of sea and land breezes. The daily maximum temperature exerted the strongest impact on O_3_ in Jinan and Qingdao, followed by NO_2_/NO. Li et al. (2020) also believes that high temperature is the main meteorological driver in summer. In contrast to Jinan, PM_2.5_ had a greater impact on O_3_ in Qingdao [73].

#### 3.3.3. Time Frequency and Phase Relationship between O_3_ and Other Air Pollutants

(1)**NO, NO_2_, NO_x_, and NO_2_/NO.** In the annual cycle, nitrogen oxides (including NO, NO_2_, and NO_x_) and O_3_ attained an inverse phase relationship, in which the value/phase of one parameter increased and that of the other parameter decreased (Figure 7). NO and O_3_ exhibited an inverse phase relationship with a short period (less than 14 days). This may occur as a result of NO titration reaction (R1). NO_2_ and O_3_, NO_x_ and O_3_ mostly attained a positive phase relationship in summer and an inverse phase relationship in winter in the short period. O_3_ production in winter was generally in the NO_x_-saturated regime that high NO_x_ concentration restrains O_3_ formation. Simultaneously, high NO emissions titration contributed to a reduction in O_3_ [11]. In summer, O_3_ was in the NO_x_-limited regime that the increase of NO_2_ was conducive to O_3_ formation (R2). Xia et al. (2021) similarly showed that the daily O_3_ concentration is higher when the daily NO_2_ concentration is higher in summer [36]. In addition to NO_x_, VOCs are also important precursors for O_3_ formation, which readily affect the variation of O_3_ concentration in winter [74,75,76]. However, VOCs were not regarded as an influencing factor in this paper. The unexplained part of the figure may be mainly attributed to VOCs.

NO + O_3_—→NO_2_ R1

NO_2_ + O_2_ + hv—→NO + O_3_ R2

In order to further clarify the relationship between NO_x_ and O_3_, wavelet coherence analysis of NO_2_/NO and O_3_ was performed (Figure 7). As can be seen from the Table 2 and Figure 7, the coherence between NO_2_/NO and O_3_ was stronger than that of NO, NO_2_, and NO_x_. There was an obvious annual cycle between the NO_2_/NO ratio and O_3_. In the annual cycle, the NO_2_/NO ratio and O_3_ concentration in Jinan changed at the same time, while the arrow in Qingdao gradually tilted downward, indicating that the NO_2_/NO ratio had fluctuated before O_3_ in Qingdao since 2018. In the short cycle, NO_2_/NO ratio had strong coherence with O_3_ from May to July, and both increased simultaneously. This is consistent with the findings of Shao et al. (2009) and Wang et al. (2019) that the O_3_ concentration increases with the NO_2_/NO ratio [28,74]. In spring, O_3_ shifted from the NO_x_-saturated regime to the NO_x_-limited regime due to NO_x_ concentration decrease (NO_2_ gradually decreased, while NO plummeted). The magnitude of each change of NO_2_ concentration in spring and summer was greater than that of NO (Appendix A). Therefore, the increase of NO_2_ concentration not only promoted the formation of O_3_, but also led to the increase of NO_2_/NO ratio. Wang et al. (2018) believe that NO titration is generally more significant on high-O_3_ days, which further leads to an increase in the NO_2_/NO ratio [77]. Similarly, the reduction of NO_2_ concentration inhibits the formation of O_3_. During the transitional stage between the NO_x_-saturated regime and the NO_x_-limited regime (spring and autumn), the changes in NO_2_/NO ratio and O_3_ were reversed. This phenomenon was prominent in Jinan during the transition from NO_x_-saturated to NO_x_-limited (April and May), while Qingdao was more significant in the transition from NO_x_-limited to NO_x_-saturated (October). The concentration of NO_2_ decreased gradually in spring, resulting in the decrease of the NO_2_/NO ratio. O_3_ was weakened by NO_x_ limitation, and its concentration gradually increased under favorable meteorological conditions (temperature rise and light enhancement). Li et al. (2021) also reached the same conclusion that the increase in O_3_ observed in spring is driven by the reduction of NO_x_ emissions [11]. In October, the ozone concentration in Qingdao was still at a high level. Under the condition of high concentration of NO_2_, high temperature and strong radiation, the concentration of O_3_ increases. The consumption of NO_2_ reduces the NO_2_/NO ratio, so that NO_2_/NO showed the opposite relationship with O_3_.

(2)**PM_10_****and PM_2.5_.** Particulate matter (PM) (including PM_10_ and PM_2.5_) fluctuate frequently in winter and spring in Jinan and Qingdao. The phase trend of PM with O_3_ was inclined to the left during the annual cycle, indicating that PM and O_3_ exhibited an inverse phase relationship, and the change in O_3_ preceded that in PM (Figure 7). In the short term (1~30 days), PM_2.5_ and O_3_ revealed opposite phases in winter and spring but the same phase in summer. In winter and spring, the PM_2.5_ concentration was high due to the increase in energy consumption (heating). An ultrahigh PM_2.5_ concentration could lead to light radiation weakening, which could reduce O_3_ production. In addition, PM_2.5_ scavenges HO_2_ and NO_x_ radicals, resulting in O_3_ reduction [78]. O_3_ production is high in summer due to the high temperatures and strong solar irradiation. The PM concentration was low in summer. When the PM_2.5_ concentration increases to a certain extent, this could enhance the photochemical reactions of O_3_ [36]. PM_10_ attained a similar periodic relationship with O_3_. In this study, the relationship between PM and O_3_ in summer was closer in coastal areas. This is consistent with the findings of Xia et al. (2021) that the daily maximum PM_2.5_ concentration has a greater impact on the daily maximum O_3_ concentration [36]. While Hu et al. (2021) and Wang et al. (2020) both showed that PM_10_ has a much greater impact on O_3_ than PM_2.5_ due to different data accuracy and study area [14,79].

#### 3.3.4. Time Frequency and Phase Relationship between O_3_ and Meteorological Factors

(1)**Wind speed.** Short-term (1~30 days) in-phase variation between WDSP and O_3_ in Jinan mostly occurred in winter and spring (Figure 8). The low temperature and low light in winter and spring lead to less O_3_ production. The observed increase in the O_3_ concentration probably occurred due to external transport. When the wind speed is high, exogenous O_3_ may be transported. This could cause an O_3_ increase in Jinan. Affected by sea and land winds, a complex correlation existed between WDSP and O_3_ in Qingdao. Sea breezes can not only lead to the accumulation of land air pollutants but can also lead to pollutant diffusion and transport [17]. This could result in the absence of obvious short-term coherence between WDSP and O_3_. In addition to WDSP, an obvious annual cycle was observed between MXSPD and O_3_. The corresponding phase relationship trended downward, indicating that the change in O_3_ preceded that in the wind speed. High wind speeds occur in winter and spring in both Qingdao and Jinan. In contrast, the O_3_ concentration is low in winter and spring.(2)**Temperature.** TEMP and O_3_ attained a high correlation over the annual cycle. Over the annual cycle, the trend was inclined upward, indicating that O_3_ change lagged behind that in TEMP. This is because the temperature rises slowly in spring, and the O_3_ concentration gradually increases thereafter. In the short term (within 14 days), mostly the same phase was observed. This indicates that O_3_ and TEMP simultaneously increased or decreased. A high temperature and notable radiation results in a high O_3_ production rate. The time–frequency relationship between O_3_ and MAX was the same as that between O_3_ and TEMP. However, the ARsq value between O_3_ and MAX (JN: 0.6283; QD: 0.5444) was higher than that between O_3_ and TEMP (JN: 0.5037; QD: 0.5185), which is consistent with Zhao et al. (2020) [3]. This may occur because the O_3_ concentration was the highest in the afternoon, and the highest temperature throughout the day also occurred in the afternoon [3.36]. Compared with Jinan, Qingdao attained a lower coherence between O_3_ and temperature.

### 3.4. Limitations

However, the present study also suffers certain uncertainties and limitations. It is widely acknowledged that VOCs are some of the most important precursors for O_3_ generation. Unfortunately, due to the lack of available high-resolution VOC data, this study did not comprehensively evaluate the possible impact of VOCs on the spatiotemporal distribution of O_3_. The analyzed O_3_ data encompassed the average values of urban internal monitoring data according to Chinese national standard HJ 663-2013, while the obtained meteorological data included single-point data. Therefore, the spatial matching results of the meteorological and O_3_ data were not completely consistent, which could lead to deviations in the analysis results in regard to the time–frequency relationship between O_3_ and meteorological parameters.

## 4. Conclusions

The combination of MGWR and wavelet analysis is excellent in exploring the effects of O_3_ distribution from multiple perspectives. This is an innovative aspect of this study, which has not been seen in previous studies. There were significant spatiotemporal differences in the O_3_ distribution in Shandong Province. On the one hand, the O_3_ concentration increased annually from 2016 to 2019 but declined in 2020 under the influence of China’s strict O_3_ pollution cleanup policies and the global COVID-19 pandemic. On the other hand, the O_3_ concentration in inland areas (namely, central, and northwestern Shandong) was higher, whereas that in eastern coastal areas was lower. In general, PD, UR, the wind speed (WDSP and MXSDP), nitrogen oxides, and PM_2.5_ generated a greater impact on O_3_ in a coastal city (Qingdao, China) than that in an inland city (Jinan, China). Whether coastal or inland cities, the ratio of NO_2_/NO was more coherent with O_3_ than that NO, NO_2_, and NO_x_.

In summary, the reasons for the observed increase in the O_3_ concentration in winter include the following: 1. reduction in nitrogen oxides; 2. reduction in the particle concentration; 3. transport of exogenous pollutants (VOCs and O_3_) under the action of wind; 4. temperature rise. The observed O_3_ concentration increase in summer is attributed to the high temperature, PM refraction and increase in nitrogen oxides. And the reasons for the decrease in O_3_ concentration in summer include rainfall, the NO titration reaction, and reduction of NO_2_.

Nevertheless, China is located in eastern Asia and the west coast of the Pacific Ocean, with many important coastal and inland cities. Therefore, clarification of the differences in the main driving factors of O_3_ between coastal and inland cities could enhance our understanding of O_3_ pollution and improve the performance of O_3_ prediction models. In addition, it would enable the Chinese government better control O_3_ pollution. Regional O_3_ pollution is a comprehensive reflection of the environment, society, economy, and development, and strengthening the leadership role of the government is therefore of crucial importance.

## Figures and Tables

**Figure 1 ijerph-19-09687-f001:**
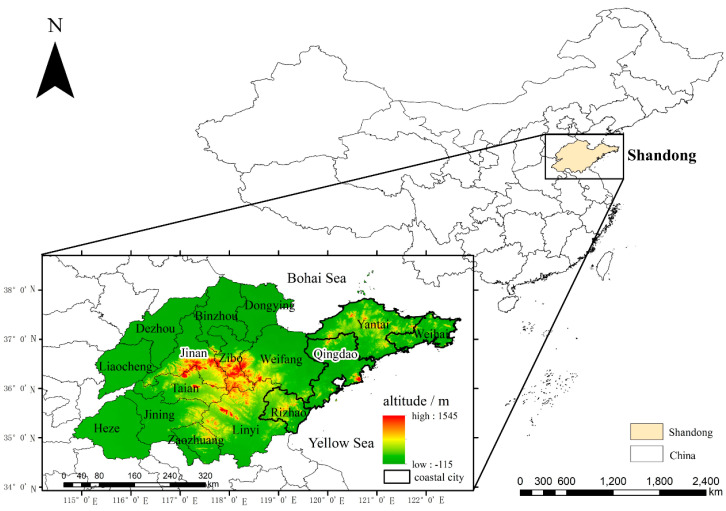
Location of the study area.

**Figure 2 ijerph-19-09687-f002:**
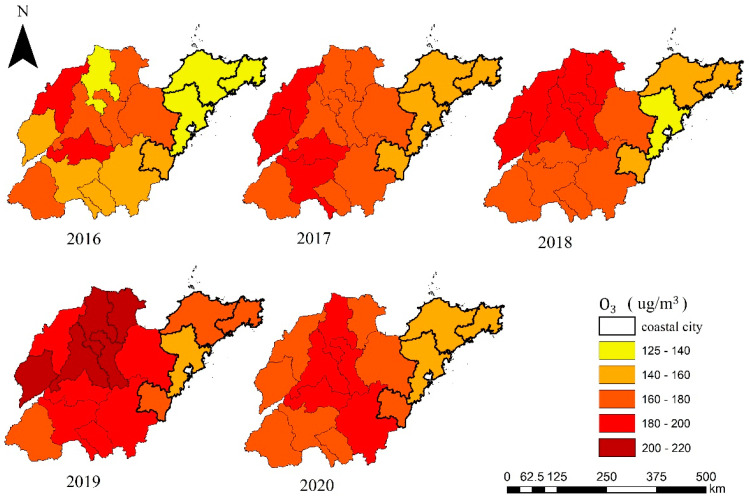
Distribution of annual O_3_ concentration in Shandong province from 2016 to 2020.

**Figure 3 ijerph-19-09687-f003:**
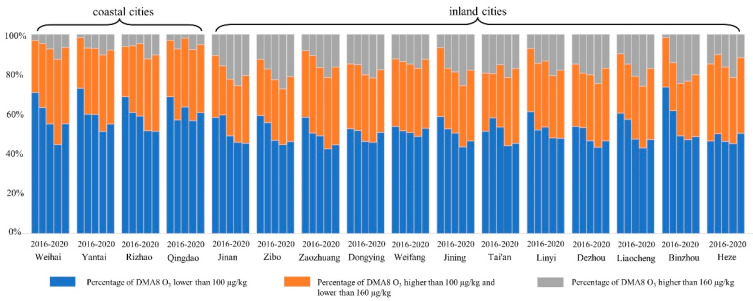
Annual change of O_3_ concentration attainment rate from 2016 to 2020.

**Figure 4 ijerph-19-09687-f004:**
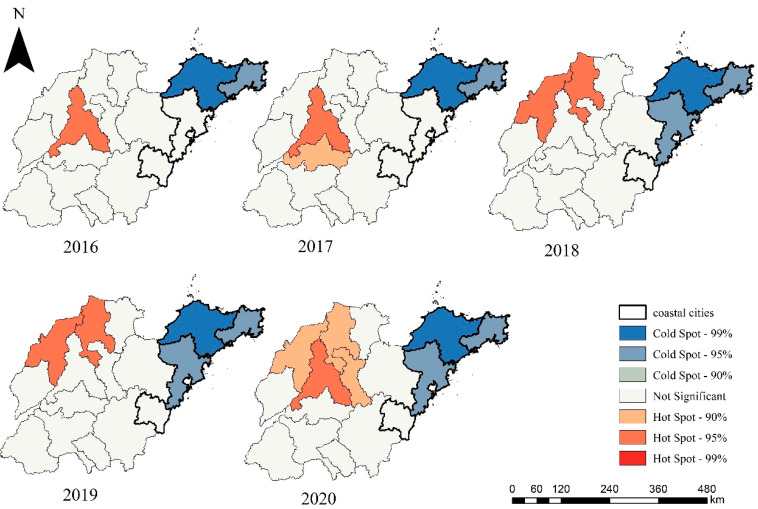
Hot spot analysis results of annual O_3_ concentrations from 2016 to 2020. Red is the high-value cluster area of O_3_, and blue is the low-value cluster area of O_3_.

**Figure 5 ijerph-19-09687-f005:**
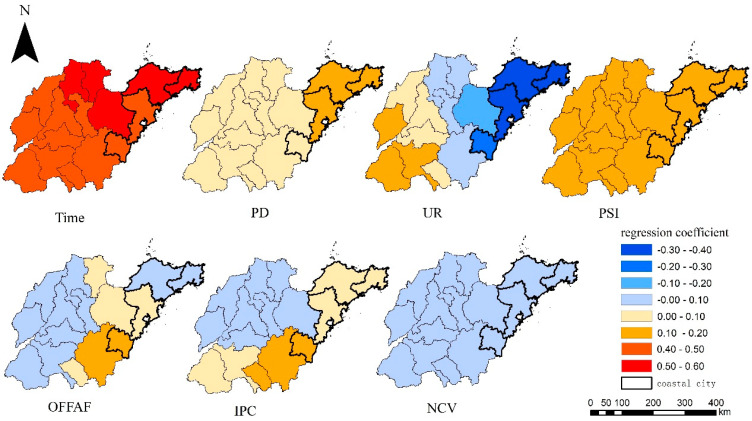
Distribution of MGWR regression coefficient.

**Figure 6 ijerph-19-09687-f006:**
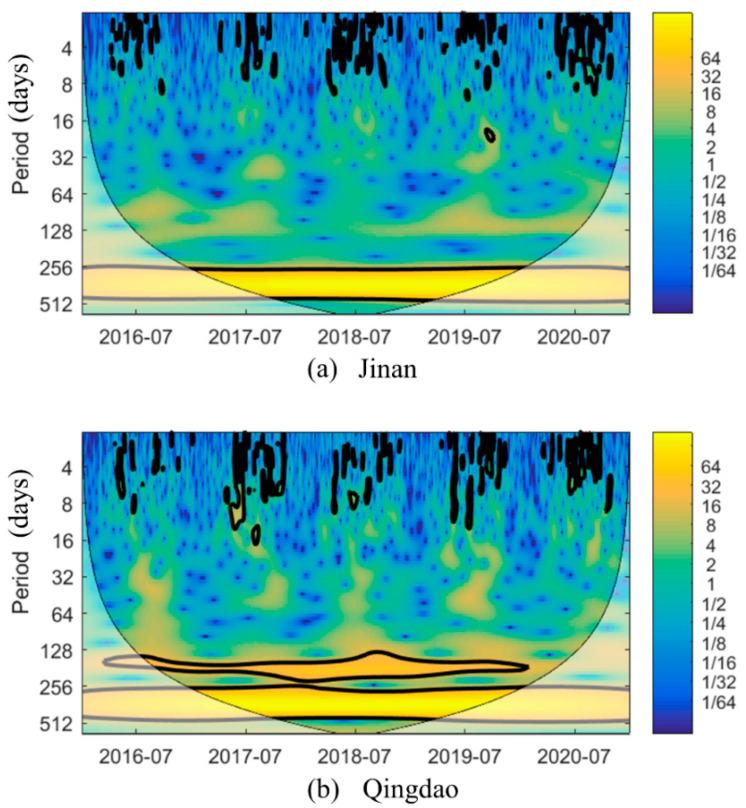
Wavelet power spectrum of O_3_. The left axis is the period (days), the bottom axis is time (days). The thick contour enclosed regions of greater than 95% confidence for a red noise process, the cone of influence where edge effects might distort the picture is shown as a lighter shade. The legend on the right shows the intensity of the wavelet power.

**Figure 7 ijerph-19-09687-f007:**
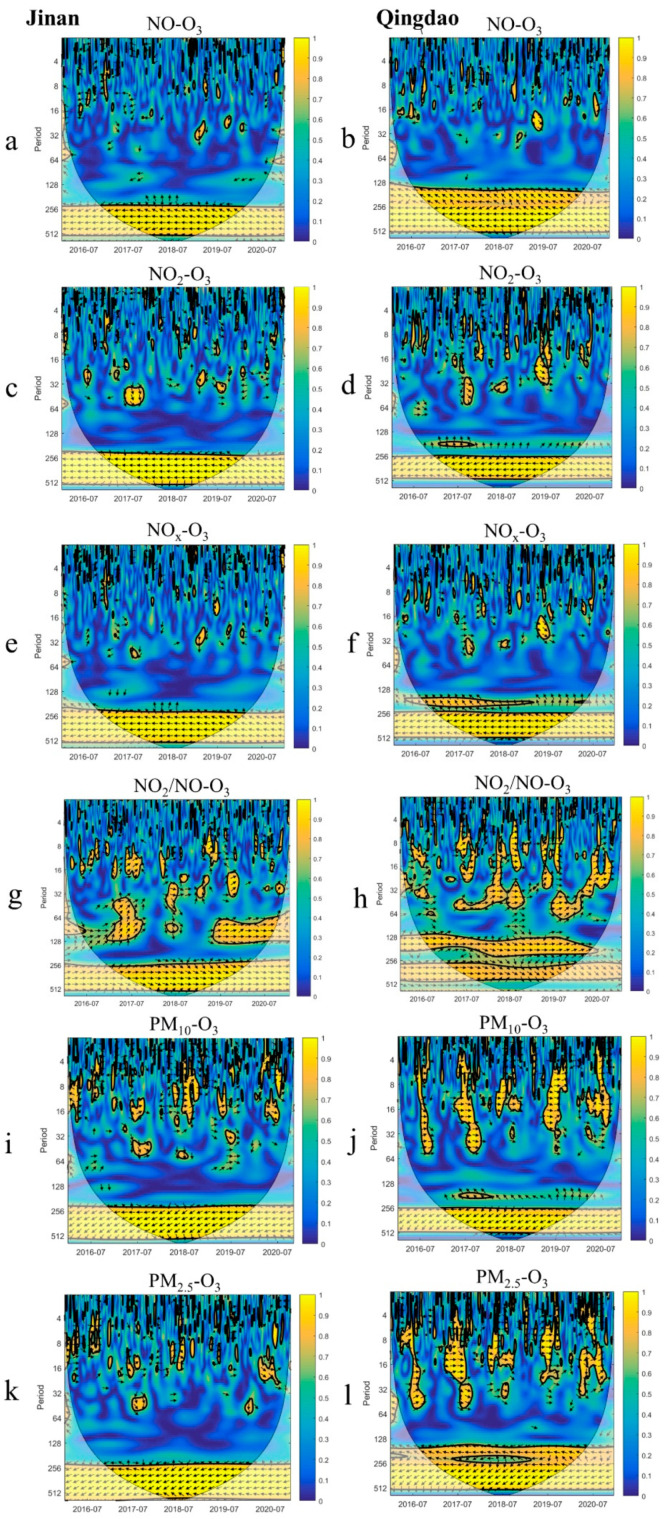
Wavelet coherence between O_3_ and NO (**a**,**b**), NO_2_ (**c**,**d**), NO_x_ (**e**,**f**), NO_2_/NO (**g**,**h**), PM_2.5_ (**i**,**j**), PM_10_ (**k**,**l**). The unit of the period is days. The thick contour enclosed regions of greater than 95% confidence for a red noise process, the cone of influence where edge effects might distort the picture is shown as a lighter shade. The legend on the right shows the intensity of the wavelet power. The relative phase relationship is shown as arrows (with in-phase pointing right, anti-phase pointing left, and O_3_ leading factors by 90° pointing straight down) [67].

**Figure 8 ijerph-19-09687-f008:**
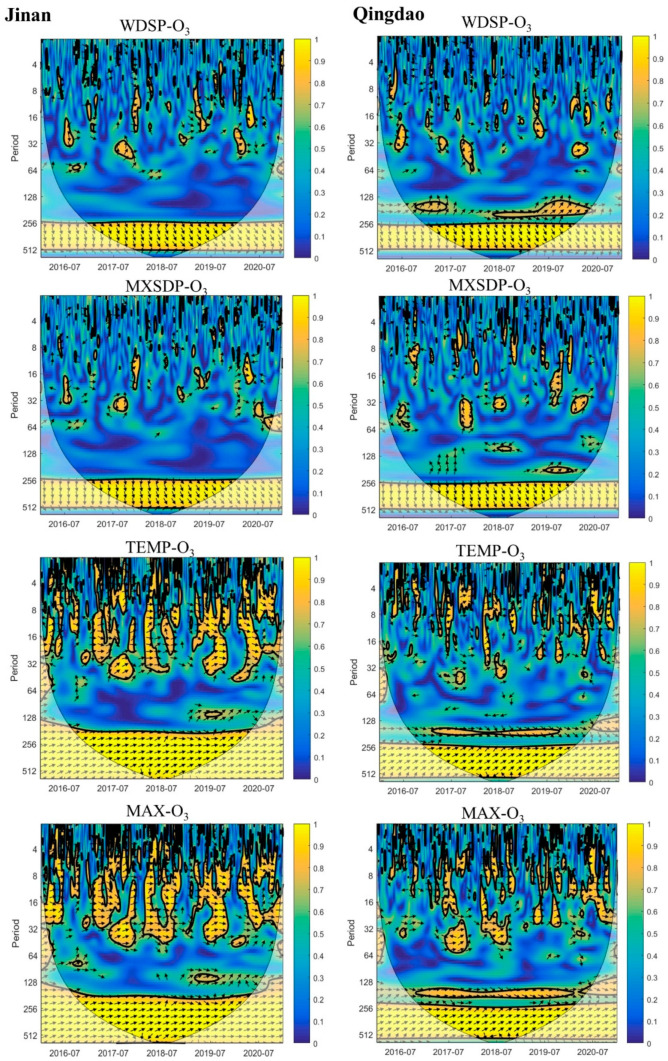
Wavelet coherence between O_3_ and meteorological factors. The unit of the period is days. The thick contour enclosed regions of greater than 95% confidence for a red noise process, the cone of influence where edge effects might distort the picture is shown as a lighter shade. The legend on the right shows the intensity of the wavelet power. The relative phase relationship is shown as arrows (with in-phase pointing right, anti-phase pointing left, and O_3_ leading factors by 90° pointing straight down) [67].

**Table 1 ijerph-19-09687-t001:** Regression results of the ordinary least squares (OLS) model, the geographically weighted regression (GWR) model, and the multiscale geographically weighted regression (MGWR) model. C is coefficient; Max is maximum value of coefficient; Mean is average value of coefficient; Min is minimum value of coefficient; AICc (Akaike information criterion) is a correction for small sample sizes, the smaller the value, the higher the goodness of fit.

Factors	OLS			GWR			MGWR			
	C	*P*	VIF	Max	Mean	Min	Max	Mean	Min	Bandwidth
Time	0.577	0.000	1.471	0.752	0.507	0.275	0.534	0.481	0.441	71
PD	0.294	0.032	1.963	0.495	0.096	−0.367	0.135	0.079	0.054	71
UR	−0.249	0.032	1.394	0.282	−0.097	−0.460	0.158	−0.065	−0.362	62
PSI	0.336	0.029	2.458	0.821	0.214	−0.142	0.149	0.135	0.125	71
OFFAF	−0.092	0.544	2.361	0.170	−0.079	−0.368	0.158	0.003	−0.081	57
IPC	0.180	0.119	1.377	0.351	0.058	−0.181	0.155	0.012	−0.064	57
NCV	−0.010	0.954	3.050	0.305	−0.001	−0.166	−0.026	−0.057	−0.087	71
AICc	218.54			180.85			171.102			
*R* ^2^	0.305			0.683			0.700			
Adj. *R*^2^	0.237			0.599			0.630			
Bandwidth				62						

**Table 2 ijerph-19-09687-t002:** ARsq (wavelet coherence coefficient) of meteorological conditions and other air pollutants with O_3_.

	NO	NO_2_	NO_x_	NO_2_/NO	PM_2.5_	PM_10_	WDSP	MXSPD	TEMP	MAX
Jinan	0.397	0.406	0.404	0.488	0.446	0.458	0.401	0.387	0.604	0.628
Qingdao	0.435	0.412	0.419	0.527	0.509	0.454	0.418	0.403	0.519	0.544

## Data Availability

The datasets were derived from the following public domain resources: http://www.cnemc.cn/ (accessed on 13 September 2021), https://www.ncei.noaa.gov/maps/global-summaries/ (accessed on 5 September 2021), http://tjj.shandong.gov.cn/tjnj/nj2021/zk/indexch.htm (in Chinese) (accessed on 3 March 2022), http://tjj.shandong.gov.cn/tjnj/nj2020/zk/indexch.htm (in Chinese) (accessed on 3 March 2022), http://tjj.shandong.gov.cn/tjnj/nj2019/indexch.htm (in Chinese) (accessed on 3 March 2022), http://tjj.shandong.gov.cn/tjnj/nj2018/indexch.htm (in Chinese) (accessed on 3 March 2022), http://tjj.shandong.gov.cn/tjnj/nj2017/indexch.htm (in Chinese) (accessed on 3 March 2022).

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
