# Peer review of "Spatiotemporal Variation in Ground Level Ozone and Its Driving Factors: A Comparative Study of Coastal and Inland Cities in Eastern China"

_ijerph, 2022, doi:10.3390/ijerph19159687_

Round 1

Reviewer 1 Report

Thanks to the respected authors for the important topic that they have addressed:

It is necessary to correct and revise the following points in the article:

1- The article must be corrected from the point of view of grammar

2- The purpose of the article should be clearly stated in the abstract and introduction

3- Questions and assumptions should be added clearly in the continuation of the introduction

4- The results of the article should be compared with the previous results

5- The applied results of this research should be mentioned at the end of the article

Good luck

Reviewer 2 Report

Review Report for International Journal of Environmental Research and Public Health

I read the whole manuscript carefully and observed the potentiality for the publication in this journal.  However, I have some serious question, which are maninly:

1.     {From 2016 to 2019, the O3 concentration in most cities increased year by year, but in 2020, the O3 concentration in all cities decreased. O3 concentration exhibited obvious regional differences, with higher levels in inland areas and lower levels in eastern coastal areas. The MGWR analysis results indicated the relationship between PD, urbanization rate (UR) and O3 was greater in coastal cities than that in the inland cities.} lines of 18 to 21, I am confused with these two sentences, and written contradictory, how O3 concentration is higher in coastal areas compared to inland by MGWR analysis?

2.     In this research, the authors have been selected two cities Qingdao from the coastal area and Jinan from the inland area to correlate the spatiotemporal distribution using MGWR analysis for the accumulation of O3.  It’s an important statistical study for this research and fully agree with authors. (merit).

3.     Fig. 2 showed the clear increment of O3 distribution in Shandong province since 2016 to 2020, over year by year.  And Fig.3 showed the annual change of O3 concentration increased from 2016 to 2020(merit).

4.     I have a benefit of doubt: Why authors have not considered Carbanaseous gases such as CO, CO2 and COx along with O3 and NOx?

On a whole, the manuscript is well written with analytical and rational examples of inland and coastal provincial cities. Mainly, authors have mentioned the limitations which are focused on VOC data. It should be improved by the authors for the future data collection.

Recommendation:

I recommend for the publication as it is.

Thank you 

Reviewer 3 Report

The differences in urban air quality between coastal and inland cities were studied with a five-year (2016–2020) dataset including air monitoring, socioeconomic statistical and meteorological data. The spatiotemporal distribution characteristics and underlying drivers of urban ozone (O3) in Shandong Province in eastern China were studied. The multiscale geographically weighted regression (MGWR) model and wavelet analysis were applied for data analyses. From 2016 to 2019, the O3 concentration in most cities increased year by year, but in 2020, the O3 concentration in all cities decreased. The regional O3 concentration differences were characterized by higher levels in inland areas and lower levels in eastern coastal areas. The maximum temperature was the most important factor influencing the O3 concentrations and the ratio of NO2 / NO was more coherent with O3. Temperature, wind speed and concentrations of nitrogen oxides and fine PM (PM2.5) influenced O3 concentrations in coastal cities more than in inland cities.

General comments

Serious air pollutions, including O3 pollution, are the motivation to study this region with typical coastal cities and typical inland cities and to study its different driving factors. It is decided to study the spatiotemporal variation characteristics of O3 concentrations by certain statistical methods. It is missing why these methods can fulfil the requirements of the objectives of this study. Also, a hypothesis of the analyses results is missing as conclusion from the overview in the Introduction.

It would be helpful if a comparison of the study results to other similar work is provided finally so that one can better understand what are the new working results. A discussion section could include these topics and would be helpful to better understand the driving factors for O3 concentration as presented shortly in the conclusion section.

The paper addresses relevant scientific questions within the scope of the journal.

The paper presents novel concepts, ideas and tools.

The scientific methods and assumptions are valid and outlined mainly so that substantial conclusions are reached.

The results are sufficient to support the interpretations.

The description of experiments and analyses is complete and precise to allow their reproduction by fellow scientists.

The quality and information of the figures and tables are fine. Figure and table captions should be more detailed to understand the content and include the abbreviations.

Title and abstract reflect the whole content of the paper. The abstract should include information about data sources.

The overall presentation is well structured and clear. The language can be polished in detail.

The mathematical symbols, abbreviations, and units are generally correctly defined and used.

Specific Comments

The authors conclude that this study could provide a scientific basis for targeted O3 pollution control in coastal and inland cities. This conclusion is more valuable if the mentioned shortcoming are corrected.

Technical corrections

The figure captions in the supplementary material are missing.

Round 2

Reviewer 3 Report

It is still missing why the selected statistical methods can fulfil the requirements of the objectives of this study. Ozone is a secondary pollutant i.e., formed by chemistry and photochemistry from precursor compounds. If MGWR is a tool for the analysis of spatial influencing factors, why it can be applied in this study? This should be discussed in section 2.

A hypothesis of the analyses results means what is expected as analyses results in relation to the overview in the Introduction. This is missing.

It would be helpful really if a comparison of the study results to other similar work is provided finally in the conclusions so that one can better understand what are the new working results.
